# Association of Elite Sports Status with Gene Variants of Peroxisome Proliferator Activated Receptors and Their Transcriptional Coactivator

**DOI:** 10.3390/ijms21010162

**Published:** 2019-12-25

**Authors:** Miroslav Petr, Agnieszka Maciejewska-Skrendo, Adam Zajac, Jakub Chycki, Petr Stastny

**Affiliations:** 1Faculty of Physical Education and Sport, Charles University, 162 52 Prague, Czech Republic; petr@ftvs.cuni.cz; 2Faulty of Physical Education, Gdansk University of Physical Education and Sport, 80-336 Gdansk, Poland; maciejewska.us@wp.pl; 3Department of Theory and Practice of Sport, The Jerzy Kukuczka Academy of Physical Education, 40-065 Katowice, Poland; a.zajac@awf.katowice.pl (A.Z.);

**Keywords:** PPAR, human performance, aerobic training, genetic predisposition, muscle fibers, anaerobic training, power, endurance training, adaptation, strength training

## Abstract

Background: Although the scientific literature regarding sports genomics has grown during the last decade, some genes, such as peroxisome proliferator activated receptors (PPARs), have not been fully described in terms of their role in achieving extraordinary sports performance. Therefore, the purpose of this systematic review was to determine which elite sports performance constraints are positively influenced by PPARs and their coactivators. Methods: The Preferred Reporting Items for Systematic Reviews and Meta-Analyses guidelines were used, with a combination of PPAR and sports keywords. Results: In total, 27 studies that referred to PPARs in elite athletes were included, where the Ala allele in *PPARG* rs1801282 was associated with strength and power elite athlete status in comparison to subelite athlete status. The C allele in *PPARA* rs4253778 was associated with soccer, and the G allele *PPARA* rs4253778 was associated with endurance elite athlete status. Other elite status endurance alleles were the Gly allele in *PPARGC1A* rs8192678 and the C allele *PPARD* rs2016520. Conclusions: PPARs can be used for estimating the potential to achieve elite status in human physical performance in strength and power, team, and aerobic sports disciplines. Carrying specific PPAR alleles can provide a partial benefit to achieving elite sports status, but does not preclude achieving elite status if they are absent.

## 1. Introduction

The scientific literature on exercise genomics has shown clear evidence that genetic markers are associated with endurance [1], power athlete status [2,3], trainability [4], and even psychological factors [5], and peroxisome proliferator activated receptors (PPARs) and/or their coactivators are often listed. While both sports performance and genomics are highly multifactorial domains, it is beneficial to summarize what phenotypic domains can be attributed to PPARs (and their coactivators) and where the analysis of phenotypic domains is redundant. Moreover, better knowledge, via functional genomics, of how PPARs (and their coactivators) may affect the individual response to physical activity or environmental factors is highly relevant not only for active individuals (athletes), but also for people who are undergoing a health treatment program that includes a physical intervention [6]. In this context, data from athletes can serve as a basis for hypotheses regarding the effectiveness of physical activity programs under extreme physiological conditions or for sedentary individuals, where the clear objective is to achieve health improvement. A recent review on the role of PPAR polymorphisms in trainability summarized several studies showing genotype/allele specific changes in health related markers [4].

PPARs are a subfamily of nuclear hormone receptors that form heterodimers with retinoid X receptors and regulate the transcription of several genes involved in lipid metabolism, energy utilization, and storage [7]. PPARs also regulate genes for glucose metabolism, carcinogenesis, and inflammation [8,9]. There are three isoforms of PPARs (PPARα, PPARβ/δ, and PPARγ, encoded by the *PPARA*, *PPARD*, and *PPARG* genes, respectively) that differ in their distribution and function [10]. For example, PPARγ is predominantly active in fat cells where it affects differentiation and growth; among other things, it is also an interesting target in pharmacotherapy for diabetes mellitus type 2 (DM2) [11]. An increased level of PPAR expression occurs in tissues that catabolize high amounts of fatty acids, such as the liver, kidney, brown adipose tissue, heart, and skeletal muscle [12,13]. In addition, muscle specific PPARβ/δ overexpression is considered to be a part of skeletal muscle plasticity. Therefore, the role of PPARs in elite aerobic performance is highly suspected [1,14].

The peroxisome proliferator activated receptor γ coactivator 1 (PGC1) family of transcriptional coactivators, consisting of three members, PGC1α, PGC1β, and the PGC-1 related coactivator (PRC), encoded by the *PPARGC1A*, *PPARGC1Β*, and *PPRC1* genes, respectively, provides important links between these transcription factors and the physiological signals controlling cellular functions related to cellular and mitochondrial energy metabolism [15,16]. PGC1α is the most frequently studied and positively regulates mitochondrial biogenesis and respiration and many other metabolic processes, including adaptive thermogenesis, gluconeogenesis, and insulin signaling [17].

The links between PPARs (and their coactivators) and muscle morphology [18], oxygen uptake [19,20], power output [21], endurance performance [18], and human trainability [4] have already been associated with elite sports status in individual studies and, in the case of *PPARGC1A* Gly428Ser, by systematic review with meta-analyses [22]. Therefore, there has been an increase of PPAR analyses in the athletic population in recent years, where *PPARA*, *PPARG*, *PPARD*, and their transcriptional coactivators’ *PPARGC1A* and *PPARGC1B* gene polymorphisms contribute to the observed phenotypes. For example, it has been shown that prolonged endurance exercise increases the transcriptional activity of *PPARGC1A* in active subjects [23]. In contrast, a recent systematic review on genes related to the level of endurance performance in mice considered at least three PPAR gene variants (and their coactivators) to be associated with endurance capacity [24]. Previous literature reviews [2,3] focused on all possible genes that might have an association with strength and power athletes’ status and suggested that PPARs have important roles that require detailed analyses. So far, only *PPARGC1A* has been reviewed in relation to power athlete status [22], and endurance athlete status and other PPARs and their coactivators have not.

Since the scientific literature in sports genomics has grown during the last decade, some genes, such as PPARs and/or their coactivators, have not been adequately described in terms of their role in athlete training and achieving extraordinary sports performance. Therefore, the purpose of this systematic review was to determine which PPARs and their coactivators are positively or negatively associated with elite sports performance constraints. We hypothesize that PPARs and/or their coactivators might determine aerobic performance and team sports elite athlete status, but not speed and strength oriented elite athlete status.

## 2. Results

The literature search resulted in a total of 4916 articles, after removing duplicates. The number of eligible articles was further reduced to 79 (including 31 reviews) after screening article titles and abstracts according to the inclusion criteria that the articles include PPARs and/or their coactivators’ gene polymorphisms at the elite athlete level (Figure 1). Of these studies, 18 were rejected following the full-text screening, and three were rejected based on the methodological quality criteria. Finally, 27 studies (Figure 1) were included in the analysis.

In total, 27 studies were included due to referring to PPARs with elite athlete status, where five PPARs were summarized as the main result of qualitative synthesis (Table 1). Thus, PPARs and their coactivators determined aerobic, speed, strength, and team sports elite athlete status. In total, 11 studies found differences between elite and subelite athletes or among elite athletes from different disciplines (Table 2). The comparison between PPARs in elite athletes and control groups only was reported in 14 studies (Table 3) and supported the main conclusions of this study. One study was a single case study and one without a control group (Table 3). The Ala allele in *PPARG* rs1801282 and the C allele in *PPARA* rs4253778 were associated with strength and power elite athlete status in comparison to subelite athletes’ status (Table 1 and Table 2). The G allele *PPARA* rs4253778, Gly allele in *PPARGC1A* rs8192678, and the *PPARD* rs2016520 C allele were associated with endurance elite athlete status in comparison to subelite athlete status (Table 1 and Table 2). The C allele in *PPARA* rs4253778 was associated with mixed endurance/strength-power (soccer) such as the *PPARD* A/C/C haplotype in rs2016520, rs2267668, and rs1053049, however only in comparison to control groups (Table 1, Table 2 and Table 3). In contrast, the G allele in *PPARA* rs4253778 was associated with mixed endurance/strength-power (soccer) with other elite athletes from combat sports and motorcycling.

## 3. Discussion

The main finding of this review was that *PPAR*s and their coactivator gene polymorphisms were related to the ability to achieve elite sports status for endurance, strength, power, and team sports oriented athletes. This consideration was specifically important for the C allele in *PPARA* rs4253778, the G allele *PPARA* rs4253778, the Gly allele in *PPARGC1A* rs8192678, and the C allele *PPARD* rs2016520, as those alleles have been found in higher frequencies in elite athletes than in subelite athletes (not just controls) and in studies including a large number of PPARs in the optimal genotype score [30,31,33] or haplotype [44]. Other findings (Table 1 and Table 3), where the genotype frequency differed between elite athletes and controls, were questionable; however, they still supported the hypothesis that PPAR alleles could influence extreme physical fitness phenotypes. Such an example was given by three studies devoted to the *PPARGC1B* gene in which two of them showed no association with athletes’ status [30,50]. However, the study of Ahmetov [18] examined the total genotype score of 15 genetic variants, where the *PPARGC1B* C allele was shown to be more common in a group of long endurance athletes compared to sedentary controls.

Although this study identified four alleles that were beneficial for elite athletes, missing the allele or the dominance of endurance or power genotypes does not mean that an athlete cannot achieve elite status, e.g., Eynon [51] reported a case study showing that athletes with the *ACTN3* R577X heterozygote variation and five out of six “endurance oriented” genotypes (including PPARs) could be successful in a long 10,000 m and short 400 m run; similarly, elite long jumpers without the power associated *ACTN3* genotype X577X have been reported [52]. On the other hand, Gonzales reported the presence of the Gly/Gly *PPARGC1A* rs8192678 genotype in a world-champion cross-country runner [38], but admitted that this genotype was not present in other elite runners. Therefore, our results can identify the potential to achieve elite sports levels, since those genotypes are also related to training response [4], but cannot play a role in whole talent identification.

The presence of the Ala allele *PPARG* rs1801282 and the C allele *PPARA* rs4253778 in elite athletes might be related to the molecular mechanisms required to sustain high anaerobic training loads [53]. Although *PPARG* rs1801282 Ala allele carriers have been found in individuals with better reactions to aerobic training in the typical population [54,55,56,57], their association in elite athletes might be related to the sustainability of periodic training, which requires tissue recovery and frequent training. Anaerobic training is accompanied by an increase in inflammatory markers, which are regulated by PPARs [8,9]. One study reported the association of the G allele in *PPARA* rs4253778 with power oriented sports (combat sports) in a comparison between elite athletes and controls (Table 2) [28], which might be explained by the mixed requirements of this sports discipline. However, this increased frequency has not been reported between elite and subelite athletes.

Most documented genetic predispositions to elite performance have been found in endurance athletes, where the G allele of *PPARA* rs4253778, the C allele of *PPARD* rs2016520, the Gly allele, and the Gly/Gly genotype of *PPARGC1A* rs8192678 have been associated with this status as candidate genes and as the crucial part of total genetic scores [30,31,33]. This confirmed the observations that the rs8192678 Gly allele may be a key element associated with the efficiency of aerobic metabolism; however, the question of how the rs8192678 Gly and Ser variants affect cardiorespiratory capacity remains unknown, although engagement of the PGC-1α coactivator in the regulation of energy metabolism, oxidative metabolism, mitochondrial biogenesis, and function has been proven, as have changes in muscle fiber types [42]. Moreover, the *PPARGC1A* rs8192678 Gly/Gly genotype has been associated with more significant increases in anaerobic threshold [54], more slow muscle fibers [55], more mitochondria activity, and a greater VO_2_ peak after aerobic training than the *PPARGC1A* rs8192678 Ser allele genotype. Another aspect is that plasmids bearing Gly or Ser at position 482 in the PGC-1α protein showed that the *PPARGC1A* 482Ser variant was less efficient as a coactivator of the myocyte enhancer factor 2C (MEF2C), which is a transcription factor regulating glucose transportation in skeletal muscle [58]. The described structure of the PPARs and their coactivators, therefore, targets many aspects necessary for elite athletic performance and might be used for training method selection or nutritional strategies [4]. Our results in the association of *PPARGC1A* Gly428Ser rs8192678 with endurance elite status seemed to be controversial with respect to previous findings [22], resulting in that this genotype was somewhat related to the power oriented athletes. This difference might be due to the contradictory finding in the original studies and that previous meta-analyses did not separate the comparisons between elite and subelite vs. comparisons between elite and control groups.

## 4. Materials and Methods

### 4.1. Review Process

The review was performed according to the Preferred Reporting Items for Systematic Reviews and Meta-Analyses (PRISMA) [59] guidelines using the review protocol assigned in PROSPERO under Database No. CRD42018082236. The final article eligibility was assessed using the adapted “Strengthening the Reporting of Observational Studies in Epidemiology” (STROBE) checklist [60] (Appendix A).

### 4.2. Literature Search

To find articles related to the role of *PPAR* polymorphisms in elite sports, we conducted a systematic computerized literature search on 20 August 2019, in PubMed (1940 to search date), Scopus (1823 to search date), and the Web of Science (1974 to search date). A combination of the following search terms was used: (PPAR) OR (peroxisome AND proliferator AND activated AND receptor) AND (sports) OR (physical AND activity) OR (endurance) OR (exercise) OR (performance) OR (movement). The search did not include comments, proceedings, editorial letters, conference abstracts, nor dissertations. Reviews were included for a manual search of their reference lists. A manual search of the reference lists of included articles was also performed (Figure 1).

### 4.3. Literature Selection

After identifying potential articles, the titles and abstracts were reviewed by two independent reviewers (P.S., M.P.) to select relevant articles for full-text screening according to the following inclusion criteria:Genotyping in PPARA, PPARG, PPARD, PPARGC1A, PPARGC1B, and genes.The population of athletes.Cross-sectional, cohort, case-control, intervention, control trial, or GWAS.

When the inclusion of articles was questionable, the reviewers agreed after a discussion. The full-text analyses of the relevant articles were performed by three independent reviewers (P.S., M.P., A.M.-S.) who also completed the data extraction form (Appendix A). During the full-text screening, the following exclusion criteria were used:(1)the full text was not available in English;(2)the study did not contain an appropriate description of athlete performance status;(3)the study did not include a specification of the selected sports discipline;(4)the study did not report PPAR frequencies for elite athletes;(5)the study was not reproducible by the methodological quality criteria.

### 4.4. Qualitative Synthesis

The result of the qualitative synthesis was based on the comparison of the type of participants in the original studies, where the highest importance was considered for comparison of elite athletes to subelite athletes. Then, the comparison between elite athletes and controls was considered as a supportive level of meaningful. Elite sports status was determined during full-text screening, where we used the original status definition of the author if it was under elite status determination [61]. The synthesis summarized three categories of sports disciplines by the dominant metabolic demand for the disciplines: strength and power oriented athletes, endurance oriented athletes, and mixed type of activity according to previous definitions [25,30,46], where team sports such as soccer or ice-hockey were considered as mixed strength-power and endurance disciplines.

## 5. Conclusions

PPARs could be used for estimating the potential to achieve elite status in human physical performance in strength, power, team, and aerobic sports disciplines. Carrying specific PPARs alleles could provide a partial benefit for achieving elite sports status, but did not preclude achieving elite status if they were absent. The Ala allele in *PPARG* rs1801282 supported the achievement of elite athlete status in strength and power disciplines. The C allele in *PPARA* rs4253778 supported the achievement of elite athlete status in mixed strength and endurance soccer, and the G allele *PPARA* rs4253778 supported achievement in endurance athletes. The Gly allele in *PPARGC1A* rs8192678 and the C allele *PPARD* rs2016520 supported the achievement of elite athlete status in endurance sports disciplines.

## Figures and Tables

**Figure 1 ijms-21-00162-f001:**
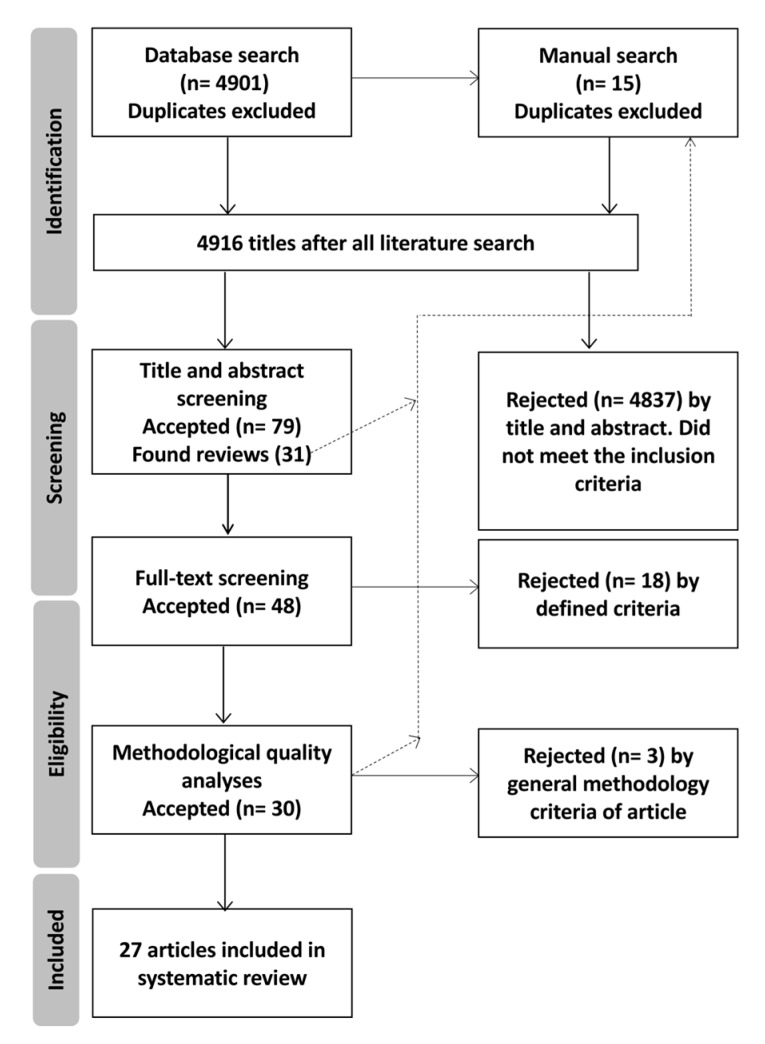
Flowchart of the review for the articles included in the tables. The dotted line demonstrates the stages where a manual search of the reference lists of the selected articles was performed.

**Table 1 ijms-21-00162-t001:** Alleles and genotypes related to elite athlete status vs. subelite status in different types of disciplines and in comparison to controls. * The minority report results specific for the reported population.

	Elite Athlete vs. Subelite Athlete	Elite Athlete vs. Controls
Strength and power oriented	*PPARG* rs1801282 Ala allele *PPARA* rs4253778 C allele	*PPARG* rs1801282 Ala allele *PPARA 7* rs4253778 C allele *PPARA*rs4253778 * GG genotype, G allele *PPARA* rs4253778 C allele *PPARGC1A* rs8192678 Gly/Gly genotype
Endurance oriented	*PPARA* rs4253778 G allele *PPARD* rs2016520 C allele *PPARGC1A* rs8192678 Gly allele, Gly/Gly genotype	*PPAR* rs4253778 * C allele *PPARD* rs2016520 C allele *PPARGC1B* rs773267 C allele *PPARA* rs4253778 GG genotype, G allele *PPARGC1A* rs8192678 Gly4 allele, Gly/Gly genotype
Mixed endurance/power		*PPARA* rs4253778 C allele, CC genotype *PPARD* haplotypes: rs2016520, rs2267668, rs1053049 *A/C/C*

**Table 2 ijms-21-00162-t002:** PPAR alleles and genotypes in elite and subelite athletes and their differences among disciplines. TGS, total genetic score.

Participant Type (*n*)	Gene/Variation	Results	Authors
Russian endurance (swimming, track-and-field, triathlon, cross-country skiing, biathlon, skating, road cycling (390), and strength oriented athletes (rowing, boxing, ice-hockey, wrestling, court tennis, weightlifting (396); controls: (1242)	*PPARA* rs4253778 (intron 7G/C)	C allele: endurance oriented < controls (*p* < 0.0001) C allele: power oriented > controls (*p* < 0.0001) CC genotype: mixed endurance/power oriented > controls (*p* = 0.0012) C allele: increasing with anaerobic component (*p* < 0.029) C allele: increasing frequency in power oriented elite athletes (*p* = 0.0316) G allele: increasing frequency in endurance oriented elite athletes (*p* < 0.0001)	Ahmetov et al., 2006 [25]
Russian elite, subelite athletes, and nonelite athletes (1539); controls (610)	*PPARD* rs2016520 (+294T/C)	C allele: athletes > controls (*p* < 0.0001) C allele: athletes in endurance oriented sports > controls (*p* < 0.0001) C allele: cyclic endurance oriented elite sports > subelite (*p* < 0.01) C allele: most pronounced between high and top level long-distance athletes (*p* = 0.013)	Ahmetov et al., 2007 [26]
Russian athletes of various strength and speed disciplines (260); controls (1073)	*PPARG* rs1801282 (Pro12Ala)	12Ala allele: athletes > controls (*p* < 0.0001) 12Ala allele: skate sprinters (*p* = 0.0002), throwers (*p* = 0.012), weightlifters (*p* = 0.003) > controls 12Ala allele: honored masters of sports > masters of sports of international rank > masters of sports > candidate masters of sports (*p* < 0.0001)	Ahmetov et al., 2008 [27]
Russian long endurance (cycling, biathlon, triathlon, long distance racing) and middle endurance (3–10 km runners, skaters, 5–10 km cross-country skiers, 800–1500 m swimmers) athletes (577); controls (1132)	*PPARA* rs4253778 (intron 7G/C) *PPARD* rs2016520 (+294T/C) *PPARG* rs1801282 (missense C/G) *PPARGC1A* rs8192678 (missense A/G) *PPARGC1B* rs7732671 (missense C/G) interactions of 10 genetic polymorphisms	C allele: long endurance athletes < non-athletes (*p* = 0.018) C allele: long endurance athletes > h (*p* = 0.006) NS A (Ser) allele: long endurance athletes < non-athletes (*p* < 0.001) C allele: long endurance athletes > non-athletes (*p* = 0.004) High number (≥9) of “endurance” alleles: long endurance elite > subelite > nonelite (*p* = 0.01) High number (≥9) of “endurance” alleles: middle endurance elite > subelite > nonelite (*p* = 0.003)	Ahmetov et al., 2009 [18]
Polish elite and subelite combat athletes (60); controls (181)	*PPARA* rs4253778 (intron 7G/C)	GG genotype: athletes > controls (*p* = 0.04) G allele: athletes > controls (*p* = 0.01)	Cieszczyk et al., 2011 [28]
Italian elite athletes (combat sports, motorcycle, soccer) (113); controls not included	*PPARA* (rs4253778) (intron 7G/C)	GG genotype: soccer > combat sports and motorcycle G allele: soccer > combat sports and motorcycle	Cocci et al., 2019 [29]
Ukrainian elite, subelite athletes, and nonelite, endurance and power oriented athletes (210); controls (326)	*PPARA* rs4253778 (intron 7G/C) *PPARG* rs1801282 (Pro12Ala) *PPARGC1B* rs7732671 (Ala2032Pro) Total genetic score of 6 gene polymorphisms	NS 12Ala allele: power oriented > endurance oriented (*p* = 0.008) NS TGS: power oriented athletes > control (*p* = 0.0142)	Drozdovska et al., 2013 [30]
Israeli national/international track-and-field athletes (155); controls 240	*PPARA* rs4253778 (intron 7G/C) *PPARGC1A* rs8192678 (Gly482Ser) *PPARD* rs2016520 (+294T/C) Total genetic score of 6 gene polymorphisms	Associated with endurance performance CI 95% Gly allele: endurance > controls (*p* < 0.05 *) Gly/Gly genotype: endurance > strength oriented > controls (*p* < 0.05 *) Associated with endurance performance CI 95%. NS TGS: endurance athletes > control and power athletes (*p* < 0.001) elite status NS	Eynon et al., 2011 [31]
Israeli track-and-field athletes (155); controls (240)	*PPARA* rs4253778 (intron 7G/C) *PPARGC1A* rs8192678 (Gly482Ser)	NS Ser/Ser genotype: endurance athletes < sprinters (*p* = 0.016) and controls (*p* = 0.012) Gly allele, Gly/Gly genotype elite athletes > non elite (*p* = 0.02)	Eynon et al., 2010 [32]
Russian elite, subelite, and nonelite soccer players (246); controls (872)	*PPARA* rs4253778 (intron 7G/C) *PPARD* rs2016520 (T294C) *PPARG* rs1801282 (Pro12Ala) *PPARGC1A* rs8192678 (Gly482Ser) Total genetic score of 8 gene polymorphisms	CC genotype: soccer players > controls (*p* = 0.0001) C allele: soccer players > controls (*p* = 0.0007) C allele: attackers > controls (*p* < 0.0001) C allele: elite soccer players > controls (*p* = 0.007) NS NS NS TGS: elite soccer players > subelite > nonelite (*p* = 0.002) TGS: elite soccer goalkeepers and midfielders > subelite > nonelite (*p* = 0.002)	Egorova et al., 2013 [33]
Lithuanian athletes, endurance (biathlon, pentathlon, road cycling, cross-country skiing, swimming, rowing, track-and-field long distance) power (weightlifting, track-and-field short distance) mixed (tennis, handball, boxing, wrestling, football) (193); controls (250)	*PPARGC1A* rs8192678 (Gly482Ser) *PPARA* rs4253778 (intron 7G/C)	Gly/Gly < Ser/Ser genotypes: anaerobic alactic maximum power (AAMP) in endurance and power athletes (*p* = 0.024) C allele: athletes > controls (*p* = 0.046) CC genotype: nonelite < subelite < elite	Gineviciene et al., 2011 [34]

The * meaning is: only minimal required significance reported.

**Table 3 ijms-21-00162-t003:** Results for PPAR alleles and genotypes in elite athletes and controls.

Participant Type (*n*)	Gene/Variation	Results	Authors
Israeli track-and-field athletes (155); controls (240)	*PPARA* rs135539 (intron 1A/C)	NS	Eynon et al., 2011 [35]
Lithuanian professional male footballers (199); controls (167)	*PPARGC1A* rs8192678 (Gly482Ser) *PPARA* rs4253778 (intron 7C/G)	Gly/Gly genotype: forwards > controls (*p* = 0.044) GG genotype: controls > forwards (*p* = 0.034)	Gineviciene et al., 2014 [36]
Russian powerlifters, weightlifters, throwers (161); controls (1202)	*PPARGC1A* rs8192678 (Gly482Ser)	Gly/Gly genotype: powerlifters > controls (*p* = 0.002) Weightlifters and throwers no difference from controls	Gineviciene et al., 2016 [37]
African and Spanish cross-country runners of different levels, one world champion (9) (case study)	*PPARGC1A* rs8192678 (Gly482Ser)	Gly/Gly genotype: present in the world champion, but not in all of the top cross-country runners	Gonzales Freire et al. [38]
Mixed nation elite endurance triathletes (196); controls not included	*PPARGC1A* rs8192678 (Gly482Ser) Total Genetic Score of 7 gene polymorphisms	NS TGS was not significantly associated with performance time	Grealy et al., 2015 [39]
Spanish male endurance athletes (104); controls (200)	*PPARGC1A* rs8192678 (Gly482Ser)	Ser482 allele: athletes < unfit controls (*p* = 0.01)	Lucia et al., 2015 [40]
Polish rowers (55); controls (115)	*PPARA* rs4253778 (intron 7C/G)	GG genotype: elite rowers > controls (*p* = 0.04) G allele: all rowers > controls (*p* = 0.03) G allele: elite rowers > controls (*p* = 0.01)	Maciejewska et al., 2011 [41]
Polish and Russian athletes of various disciplines (1605); controls (1816)	*PPARGC1A* rs8192678 (Gly482Ser)	Ser482 allele: athletes < unfit controls (*p* = 0.0001)	Maciejewska et al., 2012 [42]
Polish athletes (endurance, strength-endurance, speed-power, sprint-strength, strength, 660); controls (684)	*PPARG* rs1801282 (Pro12Ala)	12Ala allele: strength athletes > controls (*p* = 0.0007)	Maciejewska et al., 2013 [43]
Polish athletes (endurance, strength-endurance, speed-power, sprinters, 660); controls (704)	*PPARD* rs2016529 *PPARD* rs1053049 *PPARD* rs2267668 haplotypes rs2267668/ rs2016520/rs1053049	rs2016529 CC genotype: athletes > controls (*p* < 0.00001) rs1053049 TT genotype: athletes > controls (*p* < 0.0001) NSFG haplotype A/C/C: athletes < controls (*p* < 0.000001)	Maciejewska et al., 2014 [44]
Spanish professional cyclists, Olympic-class runners, world-class rowers (141); controls (123)	*PPARGC1A* rs8192678 (Gly482Ser)	NS	Muniesa et al., 2010 [45]
Polish elite athletes of different sports disciplines: power and endurance (413); controls (451)	*PPARGC1A* rs8192678 (Gly482Ser) *PPARG* rs1801282 (Pro12Ala)	NS NS	Peplonska et al., 2017 [46]
Spanish world-class rowers (39); controls (123)	*PPARGC1A* rs8192678 (Gly482Ser)	NS	Santiago et al., 2010 [47]
Greek endurance athletes (438); controls not included	*PPARGC1A* rs8192678 (Gly482Ser) *PPARA* rs4253778 (intron 7C/G) *PPARD* rs2267668 *PPARD* rs6902123 *PPARD* rs1053049	NS NS NS NS NS	Tsianos et al., 2010 [48]
Turkish elite level endurance athletes (60); controls (110)	*PPARA* rs4253778 (intron 7C/G) *PPARGC1A* rs8192678 (Gly482Ser)	GG genotype: athletes > controls (*p* = 0.006) G allele: athletes > controls (*p* < 0.001) Gly/Gly genotype: athletes < controls (*p* < 0.001) Gly482 allele: athletes < controls (*p* < 0.001)	Tural et al., 2014 [49]
Japanese endurance track-and-field athletes (175); controls (645)	*PPARD* rs2016520 (+294T/C) *PPARGC1A* rs8192678 (Gly482Ser) *PPARGC1B* rs7732671 (Ala2032Pro) Total Genetic Score of 20 gene polymorphisms	NS NS NS NS	Yvert et al., 2016 [50]

Abbreviations: TGS, total genotype score; NS, not significant.

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
