# Peer review of "Association of Elite Sports Status with Gene Variants of Peroxisome Proliferator Activated Receptors and Their Transcriptional Coactivator"

_ijms, 2019, doi:10.3390/ijms21010162_

Round 1
Reviewer 1 Report
The authors reviewed the relationship between human physical performance and variants of PPARs using the reports. The results are interesting, but the current review is not fitted to the scope of this journal. I recommend to submit to more specific journal. There are concerns that should be addressed.
1. The abbreviated words should be written by the full name when first appeared.
2. Reference 4 and 19 are the same.
3. Some grammatical errors were found, English should be improved.
Author Response
The authors reviewed the relationship between human physical performance and variants of PPARs using the reports. The results are interesting, but the current review is not fitted to the scope of this journal. I recommend to submit to more specific journal. There are concerns that should be addressed.
Answer: We agree that our topic is not best match of the IJMS journal scope in general, however we were invited to this special issue and our topic has been confirm for suitability before submission (it is in planed topic for this special issue). Our topic is closely related to the metabolic regulation and health aspect of long term movement activities. Therefore, we believe that this approach have the connection for special issue and fulfill the special issue goal.
“How key roles of PPARs in healthy and diseased organisms can be modulated to maintain or improve the optimal health of individuals and populations is of foremost interest.”
In reaction to this comment we now highlighted more the connection of genes related to sport status to metabolic regulation and health in introduction.
The abbreviated words should be written by the full name when first appeared.Answer: We amended the abbreviation appearance in the abstract and doublecheck this issue throughout the manuscript.
Reference 4 and 19 are the same.Answer: thank you for catching this detail, we reduce this reference into one appearance.
Some grammatical errors were found, English should be improved.Answer: We used the professional native speaker corrector for this resubmission.
Reviewer 2 Report
ijms-634848 Accepted Oct 31 2019
Phuntila Tharabenjasin Due Nov 08 2019
Submitted Nov 07 2019
Elite sport status associations with peroxisome 2 proliferator-activated receptors (PPARs) and their 3 transcriptional coactivator´s gene variants
Comments:
Some of the keywords include VO2max, VO2peak and mitochondrial activity.
Please clarify their connection to the systematic review
Line 48: heart [10] (X), what is (X)?
Line 58: partial review [20, 21]. How are refs 20 and 21 partial reviews? Ref 21 is a pre-print and may be unnecessary.
Line 70: which elite sports performance constrains are positively or negatively influenced by PPARs Sentence is not clear, may mention the word, “association”
Line 79-81 explain Fig 2. What do the dotted lines in Fig 2 stand for? Please explain these.
Line 96-97: Did the authors define elite and sub-elite? What do the authors mean by “cutting”? Why is “soccer” mentioned in this table, but not the other sports? What makes “soccer” special? The authors labeled this as “Figure 1”; it looks like and should be a table.
Line 100: Provide a column for elite and sub-elite
Line 101: “Table 1” The placing here looks like it is a legend not a caption
Line 102: NSGF = not significant Will NS suffice? NSGF does not seem parsimonious
Figure 2. Choice of words in the boxes of the flowchart are not suitable and should be changed (e.g. rejected and exclusion in one box is redundant; “27 included in Table 1”
--- 27 what? Table 1 should be “systematic review”
Line 205: JT wrote the paper Who is JT?; he is not in the author roster
Line 207: Not complete
Reference section: most of the years are in bold, others are not, please be consistent
S2 Table:
Egorova et al. 2013 [8] 2014 not 2013
Gonzales Freire et al. [15] Lacks year
Lucia et al. 2015 [17] 2005 or 1985? Ref 17 has these two years
The latter part of S2 Table (following page) is a repeat and should be re-examined
Author Response
Thank you for your very detailed revision of our manuscript, we think that we sufficiently improved the manuscript according to your comments.
Comments:
Some of the keywords include VO2max, VO2peak and mitochondrial activity. Please clarify their connection to the systematic review.
Answer: In general context they are less appropriate than others, thus we change them for more general such as endurance and adaptation.
Line 48: heart [10] (X), what is (X)?
Answer: This “X” remain accidentally to place reference. Now we deleted this mark.
Line 58: partial review [20, 21]. How are refs 20 and 21 partial reviews? Ref 21 is a pre-print and may be unnecessary.
Answer: We deleted the pre-print from references. In first submission, the partial review mean partial out of from PPAR´s , where referred review report specifically just PPARGC1A Gly428Ser (rs8192678). We specified this detail now in the text.
Line 70: which elite sports performance constrains are positively or negatively influenced by PPARs Sentence is not clear, may mention the word, “association”
Answer: We improve the clarity of this sentence using “association”.
Line 79-81 explain Fig 2. What do the dotted lines in Fig 2 stand for? Please explain these.
Answer: We now added the specification that the dot line demonstrate the stages where hand search has been applied in the reference list of selected articles.
Line 96-97: Did the authors define elite and sub-elite? What do the authors mean by “cutting”? Why is “soccer” mentioned in this table, but not the other sports? What makes “soccer” special? The authors labeled this as “Figure 1”; it looks like and should be a table.
Answer: The elite status definition has been used from original study definition, if in accordance to definition standards (Swan 2015) and added relevant reference. We agree that soccer is in this part little exception, therefore we remove it and we reformatted the figure to table. The cutting is not necessary as well, so we remove it. According to this changes we adopted also the text of result section.
Line 100: Provide a column for elite and sub-elite
Answer: Thank you for this comment. After careful consideration, we split the table into two, where Table 1 refer studies showing differences between elite and sub-elite or differences between different elite athlete groups.
Line 101: “Table 1” The placing here looks like it is a legend not a caption
Answer: We have moved the caption before the table.
Line 102: NSGF = not significant Will NS suffice? NSGF does not seem parsimonious
Answer: We agree and used only NS now.
Figure 2. Choice of words in the boxes of the flowchart are not suitable and should be changed (e.g. rejected and exclusion in one box is redundant; “27 included in Table 1”
--- 27 what? Table 1 should be “systematic review”
Answer: Thank you for pointing to this issue. We improved and clarified the Figure 2 according to your suggestion.
Line 205: JT wrote the paper Who is JT?; he is not in the author roster
Answer: JT was actually written accidentally, however we now included one more author Jakob Chycki, who revised the manuscript and wrote sections due to the reviewers response.
Line 207: Not complete
Answer: We have completed the list of abbreviation.
Reference section: most of the years are in bold, others are not, please be consistent.
Answer: Corrected
S2 Table:
Egorova et al. 2013 [8] 2014 not 2013
Gonzales Freire et al. [15] Lacks year
Lucia et al. 2015 [17] 2005 or 1985? Ref 17 has these two years
The latter part of S2 Table (following page) is a repeat and should be re-examined
Answer: Thank you for this very precise point. We have doublechecked and corrected all references in Table S2.
Reviewer 3 Report
Review of Elite sport status associations with peroxisome proliferator-activated receptors (PPARs) and their transcriptional coactivator ́s gene variants
. Outside: Miroslav Petr, Agnieszka Maciejewska-Skrendo, Adam Zajac2 and Petr Stastny
This review focuses on the different polymorphisms of PPARs discovered since 2005 and their association with different physiological states. In addition, the polymorphism of PPAR transcriptional co-activators is discussed. Only one table is presented with 27 articles. For the reader, questions may arise.
Major comments:
The novelty of this review on this subject is not well presented by the authors. Two very recent reviews by one of the major authors (see ref. 2 and 3 with Maciejewska-Skrendo) are cited but develop this new review compared to the two previous ones? Methodological quality is cited as a selection criterion by the authors in their methods, but how to explain that the Tsianos et al, the control group has often moved from healthy people to elite sports of different disciplines. This specific point deserves further consideration. Do we have a more in-depth analysis with more subgroups of the population, from non-sport to sport? The 18 rejected articles are based on exclusion criteria, which could be useful when a meta-analysis is performed. This article suffers from this lack of analysis. Since 9 insignificant results (NGSF) are presented, why was a meta-analysis not done? Is the meaning of the associations reported in Table 1 sufficient to be relevant 0.012-0.044 for example?
Author Response
This review focuses on the different polymorphisms of PPARs discovered since 2005 and their association with different physiological states. In addition, the polymorphism of PPAR transcriptional co-activators is discussed. Only one table is presented with 27 articles. For the reader, questions may arise.
Answer: We prepared three tables now; these present overall and partial results in comparisons between elite/sub-elite athletes or elite athletes of different sport disciplines.
Major comments:
The novelty of this review on this subject is not well presented by the authors. Two very recent reviews by one of the major authors (see ref. 2 and 3 with Maciejewska-Skrendo) are cited but develop this new review compared to the two previous ones?
Answer: There are couple needs and novelty of our review, which are now more explicitly stated in the introduction. First of all, the previous Maciejewska-Skrendo book chapters are not systematic reviews but literature reviews. Moreover, the previous review is not specifically focused on PPARs but all genes, which unable to focus on details like referring the genotype and allele results, sub-elite condition, etc. However, those specifications have been stated in the introduction now.
Methodological quality is cited as a selection criterion by the authors in their methods, but how to explain that the Tsianos et al, the control group has often moved from healthy people to elite sports of different disciplines. This specific point deserves further consideration.
Answer: We estimate the general methodological quality of Tsianos et al study and we agree that their approach is questionable, however the general methodological quality estimate whether they described the procedures in the way to allowed reproducibility and enough participants detail. Then we consider that they did. This study also did not find any significant results, which might be due to their unusual approach and does not bring methodological questions to our other results.
Do we have a more in-depth analysis with more subgroups of the population, from non-sport to sport? The 18 rejected articles are based on exclusion criteria, which could be useful when a meta-analysis is performed. This article suffers from this lack of analysis. Since 9 insignificant results (NGSF) are presented, why was a meta-analysis not done?
Answer: The meta-analysis is not possible since the raw data and population are not homogenous enough. We found relatively inconsistency in the research question among studies, which is the first assumption for meta-analyses. Therefore, we still think that meta-analyses would bring too many methodological errors.
Is the meaning of the associations reported in Table 1 sufficient to be relevant 0.012-0.044, for example?
Answer: We believe that it is relevant since the results at “borderline” in one study has been reported with higher significance in another study. Moreover, we respected the methodological and statistical approach (significance 0.05) defined by original authors. We want to noted that 0.05 is acceptable for sport science-oriented study.
Round 2
Reviewer 1 Report
My concerns were met. The manuscript was improved. However, the title had better improved. “Association of elite sport status with gene variants of peroxisome proliferator-activated receptors and their transcriptional coactivators” is better. Anyway, present title should be changed.
Author Response
My concerns were met. The manuscript was improved. However, the title had better improved. “Association of elite sport status with gene variants of peroxisome proliferator-activated receptors and their transcriptional coactivators” is better. Anyway, present title should be changed.
Answer: Thank you for the suggestion of title improvement. We change the title in this submission.
Reviewer 2 Report
December 14, 2019
ijms-634848
December 14, 2019
ijms-634848
Elite sport status associations with peroxisome proliferator-activated receptors (PPARs) and their transcriptional coactivator´s gene variants
Comments:
This is the second round of reviewing the above manuscript. Suggestions for change and improvement were expressed in the first round. The authors have responded to the comments but serious errors in the revised manuscript persist.
Previous comment: Line 205: JT wrote the paper Who is JT?; he is not in the author roster
Answer: JT was actually written accidentally, however we now included one more author Jakob Chycki, who revised the manuscript and wrote sections due to the reviewers response.
Present comment: The author answer to the previous comment quoted above contradicts with the revised version as Jacob Chycki does not appear in the first page of the manuscript as co-author.
Additional comments regarding errors in the revise manuscript:
Figure 1. The flow chart in the box of title and abstract screening. Accepted (n =79), Reviews (31) should not be separated. It is not clear that review (31) is included in accepted n = 79. In addition, the author should put the date for selection of included studies
Why did this study focus only on soccer as a team sport? It should be included in the category of endurance and / or power sport type. This study aims to determine the association of PPARs gene with elite sport status. Sport types were divided into 2 major categories. How can the author conclude the association of PPARA C allele with soccer, which was not systematically related to other polymorphisms? Should the author opt to retain this hypothesis, they should provide more information about team sports and define how it is related to PPAR in the introduction section.
Line 73-74: PPARGC1A rs8192678 in ref. 22 is not only a systematic review. It is a meta-analysis, which provides a calculated summary effect. That the results of ref. 22 contrast with the conclusion of this systematic review, should be detail in the discussion section.
The presenting of the results in Table 1 and 2 is difficult to understand. The author should decolumn the P-values. The results in the table are an heterogeneous description that include mixes between sport types (endurance / power) and specific sport type (eg. rowing, power lifter, skate, sprinter). The authors should rearrange the designation of content to present the results.
PPARGC1B is one of candidate gene in this study. The authors should explain why this gene was not included in the discussion section.
Table 1 shows that Gly/Gly genotype PPARGC1A is associated with both strength and power as well as endurance. However, it seems discrepant that the authors conclude this genotype is related only with elite endurance status. What happened to strength and power?
S2 Table should provide the legends.
Author Response
Comments:
This is the second round of reviewing the above manuscript. Suggestions for change and improvement were expressed in the first round. The authors have responded to the comments, but severe errors in the revised manuscript persist.
Answer: Thanks again for your time to spend on improving our manuscript. We improved the manuscript again and our answers are below. We believe that some comments were due to the lack of explanation of our approach to results interpretations, which we now highlighted more in the methods.
Previous comment: Line 205: JT wrote the paper Who is JT?; he is not in the author roster
Answer: JT was actually written accidentally; however, we now included one more author Jakob Chycki, who revised the manuscript and wrote sections due to the reviewer´s response.
Present comment: The author's answer to the previous comment quoted above contradicts the revised version as Jacob Chycki does not appear on the first page of the manuscript as co-author.
Answer: Thanks for this point; we included Jakub Chycki on the first page of the manuscript now (before we just did it in the cover letter because there was no possibility to make it in submission software.
Additional comments regarding errors in the revised manuscript:
Figure 1. The flow chart in the box of title and abstract screening. Accepted (n =79), Reviews (31) should not be separated. It is not clear that review (31) is included in accepted n = 79. In addition, the author should put the date for selection of included studies.
Answer: Those 31 reviews are part of 79 accepted articles after the title and abstract screening, which we clearly described in the methods section. Also, the dotted line is described in the figure legend that this part was the point of a manual search. To increase the clarity of the schedule, we now started the dotted line from the bracket with 31 reviews, so we believe that this confusion is now resolved. The reviews were the object of manual search, so that is the reason to show this number.
In terms of date for articles selection, we are really not sure about the usefulness of this information, which is not according to PRISMA. We provide the most key date – “the date of main search”. Than, some articles were included by the match of reviewers record and some during the reviewer discussion, which resulted in additional manual search in studies included after reviewer discussion. So, we are not sure what date you ask for.
Why did this study focus only on soccer as a team sport? It should be included in the category of endurance and power sport type. This study aims to determine the association of PPARs gene with elite sport status. Sport types were divided into two major categories. How can the author conclude the association of PPARA C allele with soccer, which was not systematically related to other polymorphisms? Should the author opt to retain this hypothesis, they should provide more information about team sports and define how it is related to PPAR in the introduction section.
Answer: We agree that soccer should be the category of endurance and power sport type. Therefore, we state this in method section and now is soccer included in mix type disciplines where this description is clearly stated in the method section.
Line 73-74: PPARGC1A rs8192678 in ref. 22 is not only a systematic review. It is a meta-analysis, which provides a calculated summary effect. That the result of ref. 22 contrast with the conclusion of this systematic review should be detail in the discussion section.
Answer: We now specified that reference 22 is a systematic review with meta-analyses. The contrast with the conclusion of reference 22 is now added to the discussion.
The presenting of the results in Tables 1 and 2 is difficult to understand. The author should decolumn P-values. The results in the table are a heterogeneous description that includes mixes between sport types (endurance/power) and specific sport type (eg. rowing, powerlifter, skate, sprinter). The authors should rearrange the designation of content to present the results.
Answer: In results presentation, we are using preferably summarized Table 1 which summarize Table 2 and 3. Tables 2 and 3 show purposely detailed description of variations in sport disciplines from included studies to provide those included data. We do not understand what you mean by “decolumn p values”, those are a direct level of significance, so they should be there. If you mean that p values should have our column, we found that this version would not improve the readability of the Table as other reviewers found that easy to understand as well. We also more specified the title of Table 1.
PPARGC1B is one of the candidate genes in this study. The authors should explain why this gene was not included in the discussion section.
Answer: That was because the strength and power association has been found only between elite and controls, but not between elite and sub-elite athletes. Now we mentioned PPARGC1B in discussion “Only three studies devoted to the PPARGC1B gene to which two of them showed no association with athletes' status (Yvert et a. 2016; Drozdovska et al. 2013). The study of Ahmetov et al. (2009) examined the TSG score of 15 genetic variants; the PPARGC1B C allele was shown to be more common in a group of long endurance athletes compared to sedentary controls.”
Table 1 shows that Gly/Gly genotype PPARGC1A is associated with both strength and power as well as endurance. However, it seems discrepant that the authors conclude that this genotype is related only with elite endurance status. What happened to strength and power?
Answer: That is because the strength and power association has been found only between elite and controls, but not between elite and sub-elite athletes. IF there were a discrepancy in the results done between elite and sub-elite and between elite and controls, we would interpret the elite vs. sub-elite results, which is now more explicitly stated in the method section. We believe that approach justifies all concerns about the interpretations. We understand that this might be confusing since the method section in IJMS is after the discussion.
S2 Table should provide the legends.
Answer: We provide a legend describing the items in the table
Reviewer 3 Report
This paper has been extensively revised. In particular the creation of two new tables clearly improved the manuscript.
Author Response
Thank you again for your time spend of improving our manuscript.